# Data-Driven Robust Optimization for Steam Systems in Ethylene Plants under Uncertainty

**Liang Zhao [1,2], Weimin Zhong [1,2] and Wenli Du [1,2,*]**

[1]  Key Laboratory of Advanced Control and Optimization for Chemical Processes, Ministry of Education, East China University of Science and Technology, Shanghai 200237, China; lzhao@ecust.edu.cn (L.Z.); wmzhong@ecust.edu.cn (W.Z.)

[2]  Shanghai Institute of Intelligent Science and Technology, Tongji University, Shanghai 200092, China

*  Correspondence: wldu@ecust.edu.cn; Tel.: +86-21-6425-2060

**Abstract:** In an ethylene plant, steam system provides shaft power to compressors and pumps and heats the process streams. Modeling and optimization of a steam system is a powerful tool to bring benefits and save energy for ethylene plants. However, the uncertainty of device efficiencies and the fluctuation of the process demands cause great difficulties to traditional mathematical programming methods, which could result in suboptimal or infeasible solution. The growing data-driven optimization approaches offer new techniques to eliminate uncertainty in the process system engineering community. A data-driven robust optimization (DDRO) methodology is proposed to deal with uncertainty in the optimization of steam system in an ethylene plant. A hybrid model of extraction–exhausting steam turbine is developed, and its coefficients are considered as uncertain parameters. A deterministic mixed integer linear programming model of the steam system is formulated based on the model of the components to minimize the operating cost of the ethylene plant. The uncertain parameter set of the proposed model is derived from the historical data, and the Dirichlet process mixture model is employed to capture the features for the construction of the uncertainty set. In combination with the derived uncertainty set, a data-driven conic quadratic mixed-integer programming model is reformulated for the optimization of the steam system under uncertainty. An actual case study is utilized to validate the performance of the proposed DDRO method.

**Keywords:** ethylene plant; steam system; data-driven robust optimization; uncertainty

## 1. Introduction

### 1.1. Background

Ethylene production supplies most of the basic raw materials for the petrochemical industry. Ethylene plants convert a wide range of hydrocarbon feedstocks to olefins, such as hydrogen, ethylene, and propylene by steam cracking. After quenching, compression, and separation, ethylene, propylene, and other olefin products with certain concentrations are manufactured for downstream chemical processes [1]. In an ethylene plant, the steam system usually contains extraction–exhausting steam turbines (EEST), backpressure steam turbines (BST), and heat exchangers, which provide shaft power to the compressor and pump and heat to the stream. Furthermore, letdown valves are used to regulate the pressure of different steam headers [2]. In an ethylene plant, the steam system consumes a large proportion of energy and incurs most of the operating cost. Thus, the operational optimization of steam systems for ethylene plants contributes to the reduction of the operating cost. Many research works have addressed the issue of the operational optimization of steam systems in recent years [2–4].

In most ethylene plants, steam turbines are the main consumer of the produced steam, and optimization of the operation of steam turbine is an important way to improve the efficiency of the

steam system [5]. The turbine hardware model, which considers load, size, and operation conditions, was proposed to optimize steam levels [6]. A previous research [5] developed a hybrid steam turbine model by combining the thermodynamic model and historical data, based on which a mixed-integer nonlinear programming (MINLP) model was formulated to minimize the operating cost of the steam system. Research [7] presented an mixed-integer linear programming (MILP) method for synthesis of utility system with fixed-process heat and power demand. Research [8] introduced the multi-period operational decisions into the utility system model. Research [4] proposed an MINLP approach for planning optimization of the utility system in a cogeneration plant by complex steam turbine decomposition.

Uncertainties, such as steam turbine efficiency and process demands, in an actual ethylene plant could result in an infeasible solution from existing models and optimization methodology. This condition may lead to the pressure fluctuation of the steam header and the low-efficiency operation of the steam turbine. Moreover, the fluctuating steam pressure may cause vibration of the compressor and consequently shuts down the ethylene plant. Therefore, uncertainties must be handled carefully to eliminate the influence of incorrect optimization decision. Optimization under uncertainty has attracted much research efforts [9]. Several mathematical programming techniques [10–13] have been presented to deal with uncertainty in process optimization. Research [13] proposed a worst-case optimization approach for real-time optimization of a gas lifted well system. Research [14] presented a fuzzy-credibility constrained programming approach for water resource planning under uncertainty. Research [15] utilized stochastic programming to handle uncertainties in the site utility system.

In recent years, data-driven optimization methodology, which bridges mathematical programming and machine learning, is a new process optimization technique [16,17]. Data-driven robust optimization (DDRO) has become a powerful tool to eliminate uncertainties [17,18]. Research [17] proposed a novel data-driven framework to improve the performance of traditional robust optimization by constructing an uncertainty set based on statistical hypothesis testing. Research [19] proposed a data-driven adaptive robust optimization method for operational optimization of an industrial steam system under uncertainty. Research [20] constructed an uncertainty set based on the support vector clustering method for the DDRO framework. Studies [21–23] proposed different data-driven adaptive robust optimization schemas by utilizing the data-driven uncertainty set and applied them for process scheduling and planning problems.

### 1.2. Significance

The shaft work of steam turbine in the steam system depends on the steam pressure of different steam headers and the power demand of the corresponding driven devices. However, the steam pressure and power demand change frequently under different operational conditions. Hence, steam turbine efficiencies are considered as uncertain parameters. In an actual ethylene plant, a huge amount of data, such as temperature, pressure, and flowrate of process streams, are stored in the database and easily downloaded to a local computer. The uncertain model parameters can be derived from the developed hybrid model and the collected process data. Considering that the probability distribution of the uncertainty is difficult to determine in advance, chance constraint programming and stochastic programming are unsuitable for this issue.

In this study, we propose a data-driven method for robust optimization of a steam system under device efficiency uncertainty. The hybrid building block models of the steam system are developed first by collaborating the process mechanism and operating data. The deterministic optimization model for the steam system is then built to minimize the operating cost. The parameters of the extraction–exhausting steam turbine model are considered as uncertainties. The uncertain parameter set is constructed by utilizing process historical data under different operating conditions of an actual ethylene plant. Dirichlet process mixture model (DPMM) is employed to extract statistical information from the data set to construct the uncertainty set. Based on the derived uncertainty set, a conic quadratic mixed-integer programming (CQMIP) model, which is a type of MINLP model, is formulated as the

counterpart of the original MILP model. We further present a case study to illustrate the effectiveness of the proposed approach.

### 1.3. Innovations

The main innovations of this article are summarized as follows:

(1)   A data-driven robust MILP model of steam systems is developed.
(2)   A robust counterpart of the proposed MILP model is derived as a CQMIP model.
(3)   The uncertain parameter set of the steam system is derived from process historical data in an actual ethylene plant.

### 1.4. Organization

The remainder of this article is organized as follows: Section 2 provides steam system description and problem statement. The optimization model of the steam system is presented in Section 3. In Section 4, the DDRO model of the steam system is formulated. A case study is employed to demonstrate the performance of the proposed method in Section 5. The conclusion is drawn in Section 6.

## 2. Steam System Description and Problem Statement

In an ethylene plant, the steam system generally provides power and heat to drive the compressors or pumps and heat the streams through the heat exchangers. Figure 1 presents a schematic of the steam system, which is composed of multiple steam headers, including super-high-pressure steam (SPS), high-pressure steam (HPS), medium-pressure steam (MPS), and low-pressure steam (LPS). Furthermore, four devices, namely, boilers, steam turbines, heat exchangers, and letdown valves, are always present in a steam system.

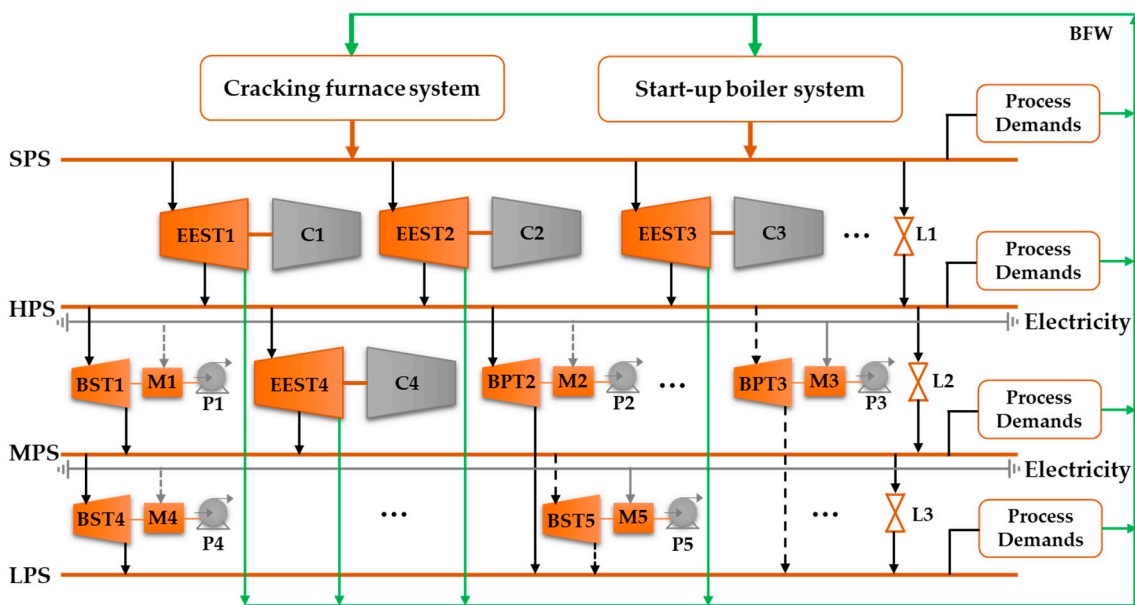

**Figure 1.** Schematic of a steam system in an ethylene plant. (EEST, extraction–exhausting steam turbine; BST, backpressure steam turbine; C, compressor; M, motor; P, pump.).

As shown in Figure 1, the SPS comes from the cracking furnace system and the start-up boiler system in a typical ethylene plant. The cracking furnace recycles heat from the high-temperature flue gas to produce SPS. The start-up boiler produces SPS by burning fuel gas to meet the process demands as a supplement. The SPS section supplies steam to the HPS section by extracting steam from the

steam turbine and letdown valve. Section MPS is supplied by sections SPS and HPS. Section LPS is provided by sections HPS and MPS. Letdown valves L1, L2, and L3 are designed to regulate the pressure of the steam headers to guarantee the safety of the steam system. The process demands mainly include streams that are heated. Steam is converted into water, which can be used as boiler feed water after purification.

In this work, we aimed to minimize the operating cost of a steam system while satisfying the process demands, such as shaft power and heat. The following assumptions are presented to develop the steam system model:

(1)    The costs of SPS from the cracking furnace and the start-up boiler are the same;
(2)    the unit costs of SPS generated by the start-up boilers under different loads are the same;
(3)    the costs of the driving pump by a steam turbine or an electrical motor are the same, and no switch cost is incurred; and
(4)    the heat demands are constant during optimization.

For a fixed shaft work demand of the compressor, different combinations of inlet and extraction steam flow rates of a steam turbine could be provided. Hence, the flow rate of the extraction steam is selected as the optimization variable. The flow rates of the inlet steam for the letdown valve is also chosen as the optimization variable to regulate the steam header pressure. Depending on the cost deviations of steam and electricity for providing per unit shaft work, the steam turbine or electrical motor is selected to drive a pump. The use of steam turbine or motor is denoted by a binary variable. The model is also subject to mass and energy balance, process demand, and variable range constraints.

The literature review in Section 1 indicate that the efficiency of steam turbines varies under different operating conditions. Therefore, the parameters of steam turbine models are considered as uncertain parameters. A DDRO paradigm is proposed for the optimization of the steam system by utilizing the hybrid model and historical data to eliminate uncertainties.

## 3. Optimization Model of Steam System in an Ethylene Plant

### 3.1. Models of Building Blocks in the Steam System

In a typical ethylene plant, the extraction–exhausting steam turbines are used to drive the cracking gas compressor and propylene compressor. These turbines consume more than half of the energy of the steam system. The letdown valve is very important to adjust the steam header pressure. In this subsection, we summarize the developed EEST models and letdown valves.

3.1.1. Steam Turbine Model

In an ethylene plant, EESTs could be decomposed into two simple steam turbines. The shaft power is specified as follows [24,25]:

$$W = W_1 + W_2 = F^{in} \cdot \left( H_1^{in} - H_1^{out} \right) + \left( F^{in} - F^{ext} \right) \cdot \left( H_2^{in} - H_2^{out} \right) \tag{1}$$

where $F^{in}$ and $F^{ext}$ are the inlet and extraction steam flow rates, respectively. $H_1^{in}$ and $H_2^{in}$ are the enthalpies of the inlet steam in the first and second stages, respectively. $H_1^{out}$ and $H_2^{out}$ are the enthalpies of the outlet steam in the first and second stages, in which $H_2^{in}$ is equal to $H_1^{out}$. Process historical data, such as pressure and temperature, are employed to calculate the enthalpy of the steam.

Based on energy and mass conservation laws, the model of EEST is presented as follows:

$$F^{in} = c^1 \cdot F^{ext} + c^2, \tag{2}$$

where $c^1 = \left( H^{ext} - H^{exh} \right) / \left( H^{in} - H^{exh} \right)$, and $c^2 = W / \left( H^{in} - H^{exh} \right)$ are the model parameters. These relationships can be derived from the process historical data of an actual plant. In this work, $c^1$ and $c^2$ are chosen as uncertain parameters.

### 3.1.2. Letdown Valve Model

Based on the process mechanism of the letdown valve, the model is developed as a linear function in Equation (3) [2].

$$F_{ld}^{out} = c_{ld} \cdot F_{ld}^{in}, \tag{3}$$

where $c_{ld} = \left(H_{ld}^{in} - H_{ld}^{water}\right)/\left(H_{ld}^{out} - H_{ld}^{water}\right)$, which is the parameter of the model. $F_{ld}^{in}$ and $F_{ld}^{out}$ are the inlet and outlet steam flow rates. $F_{ld}^{water}$ is the flow rate of boiler feed water. $H_{ld}^{in}$, $H_{ld}^{out}$, and $H_{ld}^{water}$ are the enthalpies of the inlet steam, outlet steam and boiler feed water, respectively.

### 3.2. Steam System Model for the Ethylene Plant

The optimization model of a steam system aims to minimize the total cost of all kinds of energy, including steam, electricity, and boiler feed water, and is provided as follows:

$$\min \quad \cos t = P^{SS}\sum_t F_t^{SS} + P^e \sum_m y_m^E + P^{water}\sum_{ld} F_{ld}^{water}, \tag{4}$$

where $P^{water}$, $P^E$, and $P^{SS}$ are the prices of boiler feed water, electricity, and SPS, respectively. $F_t^{SS}$ is the flow rate of the device in the steam header SPS. $F_{ld}^{water}$ is the flow rate of letdown valve *ld*. $y_m^E$ is the electricity consumption of motor *m* and is given by:

$$y_m^E = E_m \cdot y_m, \ m \in M, \tag{5}$$

where $E_m$ denotes the rated power of the motor *m*, and $y_m$ is a binary variable that indicates whether the electricity motor *m* is running or on standby.

### 3.2.1. Shaft Power Demand

The shaft power demand of a pump or compressor must be satisfied by an electrical motor or steam turbine.

$$y_t + W_t \geq W_t^D, \ \forall t \in T, \tag{6}$$

where $W_t$ and $y_t$ are the power provided for the device *t* by the steam turbine *t* or electricity motor *t*, respectively; and $W_t^D$ is the shaft power demand of device *t*.

### 3.2.2. Process Demand

The devices in each steam header *sh* must provide sufficient heat and power to meet the process demand.

$$\sum_t F_{t,sh} \cdot H_{t,sh} \geq \sum_d Q_{d,sh}, \ \forall sh \in SH, \tag{7}$$

where $F_{sh}^t$ and $H_{sh}^t$ are the flow rate and enthalpy of the device *t* in the steam header *sh*. $Q_{sh}^d$ is the of power or heat demand for device *d* in steam header *sh*.

### 3.2.3. Variable Range

The optimization variable is subject to a safety range depending on the process design and practical operating condition.

$$F^{\min} \leq F \leq F^{\max}, \tag{8}$$

where $F^{\min}$ and $F^{\max}$ are the minimum and maximum values of the variable *F*, respectively.

In conclusion, the deterministic optimization model of steam system (DOSS) in an ethylene plant is formulated as a MILP model and is expressed as follows:

$$\begin{aligned} \min \quad & cost \\ s.t. \quad & Mass\ balance\ and\ energy\ balance\ constraints\ (1)-(3) \\ & Shaft\ power\ demand\ constraint\ (6) \qquad\qquad (DOSS) \\ & Process\ demand\ constraint (7) \\ & Variables\ range\ constraint\ (8) \end{aligned}$$

## 4. DDRO Model for the Steam System in an Ethylene Plant

### 4.1. Collection of the Uncertain Parameters

We collected many process historical data from an industrial ethylene plant to construct the uncertain parameter set, which includes the entire operating period. Based on Equation (2) and the derived data set, the set of coefficients $c^1$ and $c^2$ are formed. Figure 2 presents a 3D diagram of the uncertain parameter set.

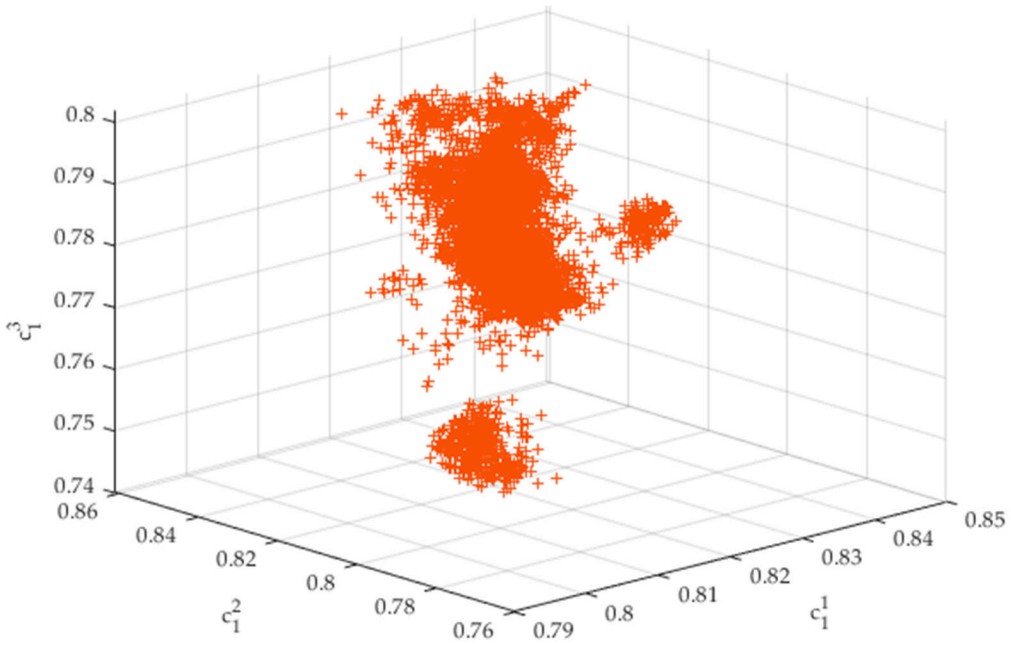

**Figure 2.** 3D diagram of the derived uncertain parameter set.

As shown in Figure 2, the parameter set has an irregular, asymmetrical, and multimode arrangement. Therefore, modelling the data by using a polynomial function is difficult. Machine learning technique is a suitable option to obtain useful information and construct a compact uncertainty set. In this study, Dirichlet process mixture model is utilized for the construction of the uncertainty set from the obtained parameter set.

### 4.2. Data-Driven Uncertainty Set Construction

Many kinds of machine learning techniques, including kernel learning [20], robust kernel density estimation [21], Dirichlet process mixture model [22], and principal component analysis [26], have been applied to capture the features of data sets for constructing an uncertainty set. According to the distribution of the parameter set, DPMM [27] was chosen to construct the uncertainty set.

The Dirichlet process mixture model is summarized as follows:

$$
\begin{aligned}
&\beta_k | \alpha \sim Beta(1, \alpha) \\
&\theta_k | F_0 \sim F_0 \\
&z_i | (\beta_1, \beta_2, \cdots) \sim Mult(\pi(\beta)) \\
&c_i | z_i \sim p(c_i | \theta_{l_i})
\end{aligned}
\tag{9}
$$

where $\beta_k$ is the mixing proportion, and $F_0$ is the prior distribution over component parameters $\theta_k$. *Mult* indicates a multinomial distribution. $c_i$ is the sample, which belongs to mixture component $z_i$.

Following Refs. [22,27], we utilized a variational inference algorithm (VIA) [28] for deriving the posterior of the DPMM. The uncertainty set is constructed as an ellipsoid and is shown as follows:

$$
C(\varepsilon) = \bigcup_{i: \gamma_i \geq \gamma^*} \{ \mathbf{c} | \mathbf{c} = \mu_i + s_i \Psi_i^{1/2} \xi, \| \xi \| \leq \Gamma_i \},
\tag{10}
$$

where $\gamma^*$ is the threshold value, and $\gamma_i$ is the weight of the component $i$. $\varepsilon$ is a predefined constraint violation degree. $\Gamma_i$ is the budget of the uncertainty set that satisfies the relationship $\Pr(\| \xi \| \leq \Gamma_i) \geq 1 - \varepsilon$, $\xi \sim t_{\omega_i + 1 - n}(0, I)$. $\mu_i$, $s_i$, and $\Psi$ are the parameters of the component $i$ derived using VIA.

*4.3. Robust CQMIP Model of the Steam System*

Considering the parameters in Equation (4) as uncertainties, the model (DOSS) is reformulated as in Equation (11) by introducing the uncertainty set (10).

$$
\min_{c \in C} \max \quad cost = P^{SS} \left( \sum_{t' \in T'} \left( c_{t'}^1 \cdot F_{t'}^{ext} + c_{t'}^2 \right) + \sum_{t'' \in T''} F_{t''}^{SS} \right) + P^e \sum_m y_m^E + P^{water} \sum_{ld} F_{ld}^{water}, \; \forall c \in C
\tag{11}
$$

where $t'$ is the index of EEST in the steam header SPS, and $t''$ is the index of other devices in the steam header SPS.

Considering the presence of uncertain parameters in the objective function, Equation (11) is rewritten into Equations (12) and (13):

$$
\min t
\tag{12}
$$

$$
P^{SS} \sum_{t'' \in T''} F_{t''}^{SS} + P^e \sum_m y_m^E + P^{water} \sum_{ld} F_{ld}^{water} + \max_{\| \xi \| \leq \Gamma_{im}} P^{SS} \left[ \mu_m z + s_m \Gamma_m \Psi_m^{1/2} z \right] \leq t, \; \forall m,
\tag{13}
$$

where $\eta_m = \left[ c_1^1, c_1^2, \cdots, c_m^1, c_m^2, \cdots \right]^T$ and $z^r = \left[ F_1^{ext}, 1, \cdots, F_m^{ext}, 1, \cdots \right]$.

The process demand constraint (7) is rewritten as follows:

$$
\sum_{t^*} H_{t^*, sh} \cdot \left( \mu_{t^*, sh}^1 z^* + s_m \Gamma_m \Psi_m^{1/2} z^* \right) + \sum_{t^{**}} F_{t^{**}, sh} \cdot H_{t^{**}, sh} \geq \sum_d Q_{sh}^d, \; \forall sh \in SH,
\tag{14}
$$

where $\eta_{t^*, sh}^1 = [c_{t^*, sh}^1, c_{t^*, sh}^2]^T$ and $z^* = [f_{t^*, sh}^{ext}, 1]$.

The variable range constraint (8) is transformed into Equations (15) and (16) as follows:

$$
\eta z^r + s_t \Gamma_t \| \Psi_t^{1/2} z^r \| \leq F^{\max},
\tag{15}
$$

$$
F^{\min} \geq \eta z^r + s_t \Gamma_t \| \Psi_t^{1/2} z^r \|,
\tag{16}
$$

where $\eta = \left[ c^1, c^2 \right]^T$ and $z^r = \left[ F^{ext}, 1 \right]$.

In summary, the DDRO model of the steam system (DDROSS) in an ethylene plant is reformulated as a CQMIP model, which is easy to be solved.

min   *t*

s.t.   *Epigraph reformulation of objective function* (13)

    *Mass balance and energy balance constraints* (1)–(3)

    *Shaft power demand constraint* (6)      (DDROSS)

    Process *demand constraints* (7), (14)

    *Variables range constraints* (8), (15), (16)

## 5. Case Study

To verify the capability of the proposed DDRO schema for steam system optimization under uncertainty, we present an actual case study from an ethylene plant. The steam system has four steam headers (SPS, HPS, MPS, and LPS), four extraction–exhausting steam turbines (EEST1–4), 29 back-pressure steam turbines (BST1–20), three letdown valves (L1–3), and more than 50 heat exchangers with fixed energy demands.

The initial conditions of the four EESTs, such as inlet and extraction steam flow rate and shaft power are summarized in Table 1. The initial conditions of BSTs and electrical motors, such as inlet flow rates, rated power, and initial states, are summarized in Figure 3. The process demands are shown in Table 2. Furthermore, the prices of SPS, electricity, and BFW are 210 CNY/t, 1.25 CNY/kwh and 10 CNY/t, respectively.

**Table 1.** Initial conditions of extracting-exhausting steam turbines.

| Steam Turbine | Inlet Flow Rate (t/h) | Extracting Flow Rate (t/h) | Shaft Power (kw) |
|---|---|---|---|
| EEST 1 | 229.6 | 182.3 | 26,512.6 |
| EEST 2 | 179.2 | 124.3 | 24,132.5 |
| EEST 3 | 69.9 | 57.0 | 18,346.3 |
| EEST 4 | 129.7 | 93.4 | 11,649.3 |

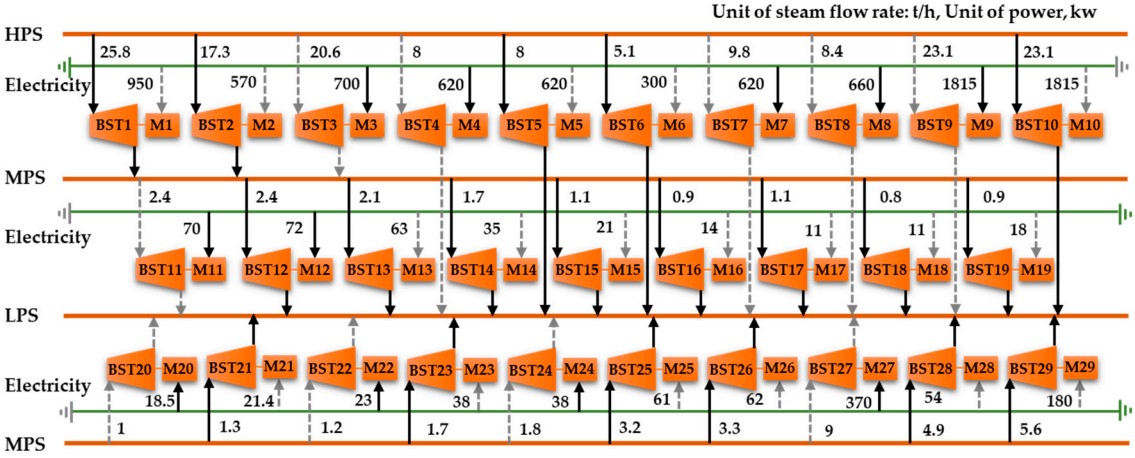

**Figure 3.** Initial conditions of backpressure steam turbines and electrical motors.

**Table 2.** Process demands of different steam headers.

| Steam Header | Process Demand (t/h) |
|---|---|
| HPS | 111.8 |
| MPS | 176 |
| LPS | 73.8 |

Moreover, the parameters of letdown valve models are as follows: $c_1 = 1.075$, $c_2 = 1.073$ and $c_3 = 1.057$. To compare the level of conservatism, we set $\varepsilon$ as 0.04, 0.1, 0.2, 0.4, and 0.8 with corresponding

budget of 4.01, 3.64, 3.32, 2.89, and 2.14, respectively. The models are coded in GAMS 24.0.2, and the DICOPT solver is used to solve the proposed DDROSS model.

The results of the deterministic model (DOSS) and the proposed DDROSS with different budgets are listed in Table 3.

**Table 3.** Results of deterministic model (DOSS) model and data-driven robust optimization model of the steam system (DDROSS) model with different budgets. CQMIP—conic quadratic mixed-integer programming.

|  | DOSS | DDROSS ($\varepsilon = 0.04$) | DDROSS ($\varepsilon = 0.1$) | DDROSS ($\varepsilon = 0.2$) | DDROSS ($\varepsilon = 0.4$) | DDROSS ($\varepsilon = 0.8$) |
|---|---|---|---|---|---|---|
| Model type | MILP | CQMIP | CQMIP | CQMIP | CQMIP | CQMIP |
| Number of constraints | 13 | 46 | 46 | 46 | 46 | 46 |
| Operating Cost (CNY/h) | 102,313 | 110,692 | 109,768 | 109,023 | 108,169 | 107,777 |

As shown in Table 3, the optimal operating cost of the DOSS model is 102,313 CNY/h. The optimized operating cost of the DDROSS model with different budgets ranges from 5.34% to 8.19%, which is higher than that of the DOSS model. The DOSS model uses a set of fixed model parameters, and the DDROSS model is subject to an uncertain parameter set. Moreover, when the value of $\varepsilon$ increases, the size of the uncertainty set and the operating cost decrease. In this case study, the difference between $\varepsilon = 0.8$ and $\varepsilon = 0.04$ is approximately 2.63%, which is used to adjust the level of the robustness and the conservatism of the DDROSS model.

The changed binary variables of DDROSS model are listed in Table 4.

**Table 4.** Changed binary variables of initial and optimized conditions.

| Electrical Motor | Initial State | Opt. State | Electrical Motor | Initial State | Opt. State |
|---|---|---|---|---|---|
| M 2 | 0 | 1 | M 17 | 0 | 1 |
| M 3 | 1 | 0 | M 18 | 0 | 1 |
| M 4 | 1 | 0 | M 19 | 0 | 1 |
| M 6 | 0 | 1 | M 21 | 0 | 1 |
| M 7 | 1 | 0 | M 22 | 1 | 0 |
| M 9 | 1 | 0 | M 23 | 0 | 1 |
| M 13 | 0 | 1 | M 25 | 0 | 1 |
| M 14 | 0 | 1 | M 26 | 0 | 1 |
| M 15 | 0 | 1 | M 28 | 0 | 1 |
| M 16 | 0 | 1 | M 29 | 0 | 1 |

Table 4 shows that more electrical motors are switched to run from standby, because the cost of electricity generation is cheaper than that of the additional steam generation. Some of the electrical motors are switched to stop from running to balance the pressure in the specific steam headers.

The extraction steam flow rates of EESTs under the initial condition, the DOSS model, and the DDROSS model with different budgets are presented in Figure 4.

Figure 4 shows that the extraction steam flow rates of the DOSS model and the DDROSS model with different budgets decrease except for EEST2 in the DDROSS model, in which $e = 0.04, 0.1$. The flow rates of extraction steam in the EESTs of DDROSS model with different budgets are greater than those of the DOSS model. The DOSS model has fixed parameters of EESTs, whereas the DDROSS model has uncertain parameters of EESTs, indicating that the latter is subject to more constraints during optimization. Depending on different efficiencies, the four EESTs have different variation trends to satisfy the shaft power and heat demands from the process.

Moreover, the steam flow rates of the letdown valves are almost zero after optimization mainly because the letdown valves are used to balance the pressure in the steam headers but do not provide power or heat to the compressor, pump, or heat exchanger. The well configuration of steam, electricity, and water can improve the efficiency and stability of the steam system.

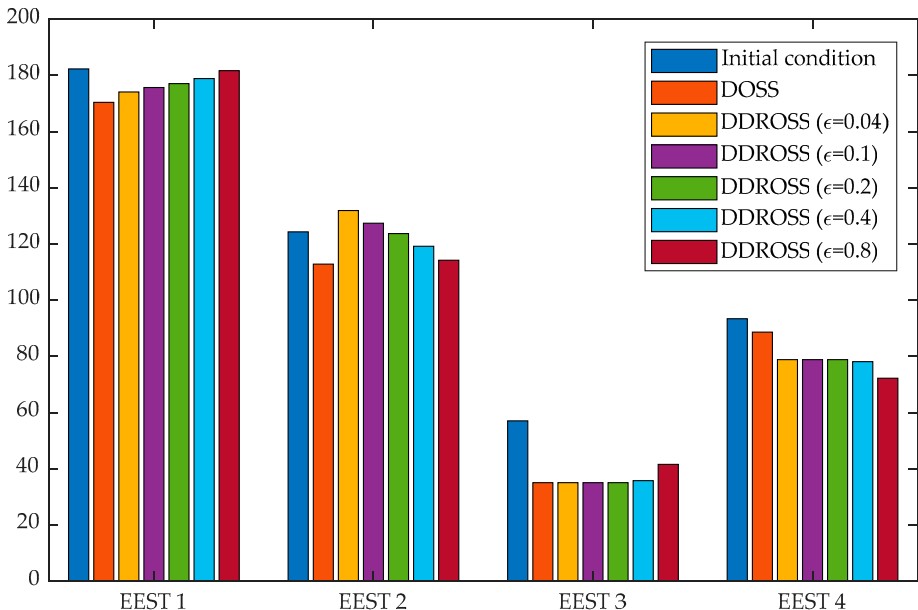

**Figure 4.** Comparison of extraction steam flow rates of EESTS (t/h) in different conditions.

To compare the performance of the proposed method and the data-driven adaptive robust optimization (DDARO) in research [19] for steam system optimization under uncertainty, we present the results in Table 5 using the same prices of the SPS, electricity, and BFW in [19].

As shown in Table 5, the numbers of continuous variables and constraints of DDROSS model are far less than those of data-driven adaptive robust counterpart for steam system (DDARCSS) model. This is because the dual transformation brings many times of auxiliary variables and constraints. The operating cost of DDARCSS model with $h = 0.05$ and $\alpha = 0.4$ is less than that of DDROSS model with different parameters due to using adaptive mechanism. Therefore, the DDROSS model is suitable for large-scale optimization problem because its numbers of variables and constraints increase linearly. On the other hand, the DDARCSS model is suitable for the small-scale problem since its numbers of variables and constraints increase exponentially. This may help us choose different robust optimization frameworks for actual optimization problems under uncertainty.

**Table 5.** Comparisons of the proposed method and data-driven adaptive robust optimization (DDARO) in research.

|  | DOSS | DDROSS ($\varepsilon = 0.04$) | DDROSS ($\varepsilon = 0.8$) | DDARCSS ($h = 0.8$, $\alpha = 0.02$) | DDARCSS ($h = 0.05$, $\alpha = 0.4$) |
|---|---|---|---|---|---|
| Model type | MILP | CQMIP | CQMIP | MILP | MILP |
| Continuous variables | 11 | 53 | 53 | 732 | 732 |
| Constraints | 13 | 54 | 54 | 369 | 369 |
| Operating Cost (CNY/h) | 88,720 | 95,779 | 93,281 | 96,366 | 89,827 |

## 6. Conclusions

This study proposes a DDRO approach for optimization of steam systems of ethylene plants under uncertainty. In the proposed method, historical operational data from an actual plant are utilized to construct an uncertainty set. The developed set covers any possible scenarios from the operating conditions, which ensures a feasible solution. The proposed DDROSS model achieves a more robust solution than the deterministic model. The level of conservatism of the DDROSS model with different budgets increases from 5.34% to 8.19%, which may help in making decision with a good trade-off between the robustness and conservatism. Future works will address how to choose the machine learning technology and different robust optimization frameworks for optimization under uncertainty.

**Author Contributions:** Methodology, W.D.; project administration, W.Z.; writing—original draft preparation, L.Z.; writing—review and editing original draft preparation, L.Z., W.Z. and W.D.; supervision, W.D.

**Funding:** This research was funded by the National Natural Science Foundation of China (Major Program: 61590923; 61873092), the International (Regional) Cooperation and Exchange Project (61720106008), the National Natural Science Fund for Distinguished Young Scholars (61725301), and the Fundamental Research Funds for the Central Universities (222201917006).

**Conflicts of Interest:** The authors declare no conflict of interest.

## Nomenclature

| | |
|---|---|
| BST | backpressure steam turbine |
| CNY | Chinese Yuan |
| Com. | Compressor |
| DDROSS | data-driven robust optimization of steam system |
| DOSS | deterministic optimization of steam system |
| EEST | extraction–exhausting steam turbine |
| HPS | high-pressure steam |
| LPS | low-pressure steam |
| M | Motor |
| MPS | medium-pressure steam |
| P | Pump |
| SPS | super-high-pressure steam |
| $c^1$ | slope of EEST model |
| $c^2$ | intercept of EEST model |
| $c_{ld}$ | slope of letdown valve model |
| $E_m$ | rated power of the motor $m$ |
| $F^{ext}$ | extraction steam flow rate of EEST |
| $F^{in}$ | inlet steam flow rate of EEST |
| $F^{max}$ | maximum value of the variable $F$ |
| $F^{min}$ | minimum value of the variable $F$ |
| $F_{ld}^{in}$ | inlet steam flow rate of letdown valve |
| $F_{ld}^{water}$ | flow rate of letdown valve $ld$ |
| $F_{sh}^{t}$ | flow rate of the device $t$ in steam header $sh$ |
| $F_{t}^{SS}$ | flow rate of the device in steam header SPS |
| $H_1^{in}$ | enthalpy of inlet steam in the first stage |
| $H_2^{in}$ | enthalpy of inlet steam in the second stage |
| $H_1^{out}$ | enthalpy of outlet steam in the first stage |
| $H_2^{out}$ | enthalpy of outlet steam in the second stage |
| $H_{ld}^{in}$ | enthalpy of inlet steam for letdown valve $ld$ |
| $H_{ld}^{out}$ | enthalpy of outlet steam for letdown valve $ld$ |
| $H_{ld}^{water}$ | enthalpy of boiler feed water for letdown valve $ld$ |
| $H_{sh}^{t}$ | enthalpy of the device $t$ in steam header $sh$ |
| $P^E$ | price of electricity |
| $P^{SS}$ | price of SPS |
| $P^{water}$ | price of boiler feed water |
| $Q_{sh}^{d}$ | power or heat demand for device $d$ in steam header $sh$ |
| W1 | First-stage shaft power of EEST |
| W2 | Second-stage shaft power of EEST |
| $W_t$ | power provided for the device $t$ by steam turbine $t$ |
| $W_t^D$ | shaft power demand of device $t$ |
| $y_m$ | binary variable to indicate that electricity motor $m$ is running or standby |
| $y_m^E$ | electricity consumption of motor $m$ |
| $y_t$ | power provided for the device $t$ by electricity motor $t$ |

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
