# Peer review of "Data-Driven Robust Optimization for Steam Systems in Ethylene Plants under Uncertainty"

_processes, doi:10.3390/pr7100744_

Round 1
Reviewer 1 Report
Comments and Suggestions for Authors
This paper illustrates a data-driven method for robust optimization of steam system under device efficiency uncertainty. However, there are serious shortcomings in the organization, structure of the content in this article.
Some of the reasons are listed below:
The structure of introduction part is very disordered. In introduction section, the research background and significance, the innovations and the structural arrangements of this paper should be summarized, explained, and given, respectively. The analysis on Part 5 Case Study lacks depth, especially the data in tables. In addition, what results can be obtained from these data? What are the differences between these results and the current research results in related fields? The conclusions should be pointed out with the major findings described. In a present version of the paper, the conclusions are too general and should be rewrite.
Author Response
Data-driven robust optimization for steam systems in ethylene plants under uncertainty
Liang Zhao, Weimin Zhong and Wenli Du
Reviewer 1
The authors are most grateful to the reviewer for the helpful comments.
Reviewer’s general comments
This paper illustrates a data-driven method for robust optimization of steam system under device efficiency uncertainty. However, there are serious shortcomings in the organization, structure of the content in this article.
Answer: We sincerely thank you for sharing the time carefully reviewing our work and providing valuable comments. In the revised manuscript, we have addressed all the comments mentioned by the reviewer. Our replies to your comments are listed point by point as follows.
The structure of introduction part is very disordered. In introduction section, the research background and significance, the innovations and the structural arrangements of this paper should be summarized, explained, and given, respectively.
Answer: Thank you very much for your comment. The introduction section is reorganized according to your suggestion. The innovations and arrangements of the paper is added in the revised version.
Action:
In Pages 1-3 of the revised manuscript:
Introduction
1.1 Background
The ethylene producing process supplies most of the basic raw materials for the petrochemical industry. Ethylene plants convert a wide range of hydrocarbon feed stocks to olefins such as hydrogen, ethylene and propylene et.al via steam cracking. After quenching, compression and separation, the ethylene, propylene and other olefin products with certain concentrations are manufactured for the downstream chemical processes [1]. In an ethylene plant, steam system usually contains extraction-exhausting steam turbine (EEST), backpressure steam turbine (BST), and heat exchanger, which provide shaft power to the compressor and pump, as well as heat to the stream. Furthermore, there are also letdown valves which regulate the pressure of different steam headers [2]. In an ethylene plant, the steam system consumes a large part of the energy and takes up most of the operating cost. Thus, the operational optimization of steam systems for ethylene plants contributes to the reduction of the operating cost. Lots of research works has addressed the issue of operational optimization of steam systems in recent years [2-4].
In most of the ethylene plants, steam turbines are the main consumer of the produced steam, optimizing the operation of steam turbine is an important way to improve the steam system efficiencies [5]. The turbine hardware model (THM), which considers load, size and operation conditions, was proposed to optimize the steam levels [6]. The research [5] developed a hybrid steam turbine model by combining the thermodynamic model and historical data, based on which a mixed-integer nonlinear programming (MINLP) model was formulated to minimize the operating cost of a steam system. The research [7] presented a mixed-integer linear programming (MILP) method for the synthesis of utility system with fixed process heat and power demand. The research [8] introduced the multi-period operational decisions into the utility system model. The research [4] proposed a MINLP approach for the planning optimization of the utility system in a cogeneration plant by the complex steam turbine decomposition method.
However, the uncertainty such as the steam turbine efficiencies and process demands in a real-world ethylene could make the existing modeling and optimization methodology result in an infeasible solution. This may lead to pressure fluctuation of steam-header and a low-efficiency operation of steam turbine. More than this, the fluctuating steam pressure may cause vibration of the compressor, which will lead to a shutdown of the ethylene plant. Therefore, the uncertainty must be handled carefully to relieve the influence of the incorrect optimization decision. Optimization under uncertainty has attracted many research efforts [9]. Many kinds of mathematical programming techniques [10-13] have been presented to deal with the uncertainty in the process optimization. The research [13] proposed a worst case optimization approach for real-time optimization of gas lifted well system. The work [14] presented a fuzzy-credibility constrained programming approach for water resources planning under uncertainty. The work [15] utilized stochastic programming to handle the uncertainties in the site utility system.
In recent years, data-driven optimization methodology, which bridges mathematical programming and machine learning, becomes a new way for process optimization [16, 17]. More recently, data-driven robust optimization (DDRO) has grown to be a powerful tool to hedge against the uncertainty [17, 18]. The research [17] proposed a novel data-driven framework to improve the performance of traditional robust optimization through constructing the uncertainty set based on statistical hypothesis testing. The research [19] proposed a data-driven adaptive robust optimization method for industrial steam system operational optimization under uncertainty. The work [20] constructed the uncertainty set based on the support vector clustering method for the data-driven robust optimization framework. The researches [21-23] studied the data-driven adaptive robust optimization schemas by utilizing the data-driven uncertainty set, and applied them to process scheduling and planning problems.
1.2 Significance
Usually, the shaft work of steam turbine in the steam system depends on the steam pressures of different steam headers and the power demand of the corresponding driven devices. However, the steam pressures and power demands change frequently under different operational conditions. So the steam turbine efficiencies are considered as uncertain parameters. In a real-world ethylene plant, huge numbers of data such as temperature, pressure and flowrate of process streams are stored in the database and easily downloaded to a local computer. The uncertain model parameters can be derived from the developed hybrid model and the collected process data. Since it is hard to know the probability distribution of the uncertainty in advance, the chance constraint programming and stochastic programming are not suitable for this issue.
In this paper, we propose a data-driven method for robust optimization of steam system under device efficiency uncertainty. The hybrid building block models of the steam system are developed firstly by collaborating the process mechanism and operating data. After that, the deterministic optimization model for the steam system is built to minimize the operating cost. The parameters of the extraction-exhausting steam turbine model are considered as uncertainties. The uncertain parameters set is constructed from process historical data of different operating conditions of a real-world ethylene plant. Dirichlet process mixture model (DPMM) is employed to extract the statistical information from the data set to construct the uncertainty set. Based on the derived uncertainty set, a conic quadratic mixed-integer programming (CQMIP) model, which is a type of MINLP model, is formulated as the counterpart of the original MILP model. We further present a case study from to illustrate the effectiveness of the proposed approach.
1.3 Innovations
The main innovations of this article are summarized as follows:
(1) A data-driven robust MILP model of steam systems is developed.
(2) The robust counterpart of the proposed MILP model is derived as CQMIP model.
(3) The uncertain parameters set of steam system is derived from process historical data in a real-world ethylene plant.
1.4 Organization
The remainder of this article is organized as follows. Section 2 provides steam system description and problem statement. The optimization model of the steam system is presented in Section 3. In Section 4, the DDRO model of the steam system is formulated. A case study is employed to demonstrate the performance of the proposed method in Section 5. The conclusion is drawn in Section 6.
The analysis on Part 5 Case Study lacks depth, especially the data in tables. In addition, what results can be obtained from these data? What are the differences between these results and the current research results in related fields?
Answer: Thanks a lot for the valuable comment. To demonstrate the results more intuitively, Tables 2 and 6 are transformed to figures. We add detailed descriptions of the results obtained from the data. The differences between these results and the related research results are also focused on in the revised manuscript.
Action:
In Pages 7-10 of the revised manuscript:
To verify the capability of the proposed DDRO schema for steam system optimization under uncertainty, we present a real-world case study from an ethylene plant. There are 4 steam headers (SPS, HPS, MPS, and LPS), 4 extraction-exhausting steam turbines (EEST1-EEST4), 29 back-pressure steam turbines (BST1 to BST20), 3 types of letdown valves (L1-L3) and more than 50 heat-exchangers with fixed energy demands in the presented steam system.
The initial conditions of the four EESTs such as inlet and extraction steam flow rates, shaft powers are given in Table 1. The initial conditions of BSTs and electrical motors such as inlet flow rates, rated power and initial states are given in Fig. 3. The process demands are shown in Table 2. Furthermore, the prices of SPS, electricity and BFW are 210 CNY/t, 1.25 CNY/kwh and 10 CNY/t, respectively (where CNY stands for “Chinese Yuan”).
Table 1. The initial conditions of extracting-exhausting steam turbines.
|
Steam turbine |
Inlet flow rate (t/h) |
Extracting flow rate (t/h) |
Shaft power (kw) |
|
EEST 1 |
229.6 |
182.3 |
26512.6 |
|
EEST 2 |
179.2 |
124.3 |
24132.5 |
|
EEST 3 |
69.9 |
57.0 |
18346.3 |
|
EEST 4 |
129.7 |
93.4 |
11649.3 |
Figure 3. The initial conditions of backpressure steam turbines and electrical motors.
Table 2. The process demands of different steam headers.
|
Steam header |
Process demand (t/h) |
|
HPS |
111.8 |
|
MPS |
176 |
|
LPS |
73.8 |
Moreover, the parameters of letdown valve models are given as follows: c1=1.075, c2=1.073 and c3=1.057. To compare the level of conservatism, the parameter is set as 0.04, 0.1, 0.2, 0.4 and 0.8, which derives the budget that equals to 4.01, 3.64, 3.32, 2.89, and 2.14. The models are coded in GAMS 24.0.2, and the DICOPT solver is used to solve the proposed DDROSS model.
The results of the deterministic model (DOSS) and the proposed data-driven robust optimization model (DDROSS) with different budgets are listed in Table 3.
Table 3. The initial conditions of extracting-exhausting steam turbines.
|
|
DOSS |
DDROSS ( =0.04) |
DDROSS ( =0.1) |
DDROSS ( =0.2) |
DDROSS ( =0.4) |
DDROSS ( =0.8) |
|
Model type |
MILP |
CQMIP |
CQMIP |
CQMIP |
CQMIP |
CQMIP |
|
Number of constraints |
13 |
46 |
46 |
46 |
46 |
46 |
|
Operating Cost (CNY/h) |
102,313 |
110,692 |
109,768 |
109,023 |
108,169 |
107,777 |
As seen in Table 3, the optimal operating cost of DOSS model is 102,313 CNY/h. The optimized operating costs of DDROSS model with different budgets are 5.34% to 8.19%, which higher than the DOSS model. This is because the DOSS model uses a set of fixed model parameters and the DDROSS model subjects to an uncertain parameter set. Moreover, when the value of increases, the size of uncertainty set becomes smaller and the operating cost decreases. In this case study, the difference between ε=0.8 and ε=0.04 is about 2.63%, which is used to adjust the level of the robustness and the conservatism of the DDROSS model.
The changed binary variables of DDROSS model are listed in Table 4.
Table 5. The initial conditions of backpressure steam turbines and electrical motors.
|
Electrical motor |
Initial state |
Opt. State |
Electrical motor |
Initial state |
Opt. State |
|
M 2 |
0 |
1 |
M 17 |
0 |
1 |
|
M 3 |
1 |
0 |
M 18 |
0 |
1 |
|
M 4 |
1 |
0 |
M 19 |
0 |
1 |
|
M 6 |
0 |
1 |
M 21 |
0 |
1 |
|
M 7 |
1 |
0 |
M 22 |
1 |
0 |
|
M 9 |
1 |
0 |
M 23 |
0 |
1 |
|
M 13 |
0 |
1 |
M 25 |
0 |
1 |
|
M 14 |
0 |
1 |
M 26 |
0 |
1 |
|
M 15 |
0 |
1 |
M 28 |
0 |
1 |
|
M 16 |
0 |
1 |
M 29 |
0 |
1 |
From Table 4, it is observed that more electrical motors are switched to run from standby. This is because the cost of electricity generation is cheaper than that of the additional steam generation. Some of the electrical motors are switched to stop form running to balance the pressures of the specific steam headers.
The extraction steam flow rates of EESTs in the initial condition, DOSS model and DDROSS model with different budgets are presented in Fig. 4.
Fig. 4 Comparison of extraction steam flow rates of EESTS (t/h) in different conditions.
From Fig. 4, it is observed that the extraction steam flow rates of the DOSS model and the DDROSS model with different budgets become smaller than the initial condition except EEST2 in the DDROSS model with the parameter equals to 0.04 and 0.1. The flow rates of extraction steam in EESTs of DDROSS model with different budgets are all greater than the DOSS model. This is because the DOSS model has the fixed parameters of EESTs and the DDROSS model has uncertain parameters of EESTs, which means that DDROSS model subjects to more constraints during the optimization process. Depending on the different efficiencies, the four EESTs have different variation trends to satisfy the shaft power and heat demands from the process.
Moreover, the steam flow rates of letdown valves are almost zero after optimization, mainly because the letdown valves are used to balance the pressures of steam headers, but do not provide power or heat to the compressor, pump or the heat exchanger. The well configuration of steam, electricity and water can make the steam system achieve a better and more stable state.
The conclusions should be pointed out with the major findings described. In a present version of the paper, the conclusions are too general and should be rewrite.
Answer: We sincerely thank you for this comment. The conclusion section is rewritten and the major findings of this work are emphasized in details in the revised manuscript.
Action:
In Page 10 of the revised manuscript:
This paper proposes a DDRO approach for the optimization of the steam systems of ethylene plants under uncertainty. In the proposed method, the historical operational data in a real-world plant are utilized to construct the uncertainty set. The developed set covers any possible scenarios of the operating conditions, which guarantee that the solution is feasible. The results show that the proposed DDROSS model achieved a more robust solution than the deterministic model. The level of conservatism of the DDROSS model with different budgets is changed from 5.34% to 8.19%, which may help to make a decision with a good trade-off between the robustness and conservatism. However, the proposed DDRO model is a static robust optimization problem, which always makes the solution more conservative. Future works will address this issue by using advanced machine learning technology and adaptive robust optimization framework.

Reviewer 2 Report
In this draft, the authors proposed a data-driven robust optimization framework for steam systems with uncertainties in ethylene plants. To effectively deal with the uncertainty in the optimization of the steam system, they developed a hybrid model of steam turbine as well as a deterministic mixed-integer linear programming model that aims to minimize the operating cost of the ethylene plant. Additional, Dirichlet process mixture model is introduced to capture the feature for the uncertain parameters set. Using this proposed DDRO framework, a real-world example is used to demonstrate the effectiveness and applicability.
This study is a very interesting topic. The draft is well written and organized so that it is easy to follow. The proposed DDRO framework achieves significant improvement in terms of the performance metric for the real case. The review, therefore, recommends accepting this paper with minor revision, for example, the grammar issue which may need a further check and Line 233, three weird circles?
Reviewer 3 Report
I read your manuscript. and these are my comments to improve your paper.
1. Your manuscript is not easy to catch what you want to say.
-No logical description.
-No nomenclature of your many symbols.
2. Your presentation is only presented with Fig. 2 which does not clearly explain your results.
3. You are trying to your results by numerical results on Tables. But it is not clear how to collect your results. This means that your paper is not clear to explain your simulation work.
4.You shows many equations to explain your model. Many equations are not clearly presented. Please check equations.
5. It is not correct that showing many equations is good paper.
6.Please revise with careful consideration to transfer your good wok.
7. Please check Eq. (15)-(18).
8. Please check Line 233
9. You described many constraints as numbers. Please explain what kind of constraints will be?
Reviewer 4 Report
This work addresses the optimization of steam systems in ethylene plants under uncertainty. In the proposed data-driven robust optimization approach, the uncertainty set is constructed by inferring a Dirichlet process mixture model from data. The resulting robust optimization model is then reformulated into a mixed-integer quadratic program (MICP), which can be solved using off-the-shelf MINLP solvers. A case study using real-world data is presented.
The proposed approach is sound as it is a direct application of the method developed by Ning and You (2017, AIChE Journal 63(9), 3790-3817), and it is great to see it applied to an industrial case study with real-world data like this one. My main concern is the lack of novelty in and apparent motivation for this work. The first author of this manuscript recently published a paper with Ning and You titled “Operational Optimization of Industrial Steam Systems Under Uncertainty Using Data-Driven Adaptive Robust Optimization” (Zhao et al., 2019, AIChE Journal 65(7)). Essentially the same operational steam system model is used in that paper, as well as a data-driven robust optimization approach. The three main differences that I see between the approaches presented in that paper and this manuscript are the following: (1) in that paper, the uncertainty set is constructed using robust kernel density estimation (RKDE), whereas a Dirichlet process mixture model is used here; (2) the robust optimization model in that paper is adaptive, i.e. it considers recourse, whereas the model here is static; and (3) the resulting model in Zhao et al. (2019) is an MILP, whereas the problem here is reformulated into an MIQP. This observation begs the following question: Compared to the previous paper, what is the motivation for this work? It is unclear to me what the benefit of using a Dirichlet process mixture model vs. RKDE is. It results in a harder optimization problem (MIQP vs. MILP), and the proposed model does not consider recourse, which makes it significantly more conservative than the previous model. Without further justification, this seems to be an obviously less suited approach.
I recommend reconsidering the manuscript after major revision. In the revised manuscript, please justify the choice of this approach compared to what has already been done in that very similar previous work. I would greatly appreciate a comparison between these two approaches in the case study.
Author Response
Thank you for taking time review our manuscript carefully. The comments are very constructive. We response you comments in the revised manuscript.

Round 2
Reviewer 1 Report
The authors revised the article as I suggested. This revised version can be published in Processes.
Author Response
Thank you very much for the valuable comments. Two native speakers polish the language in the revised manuscript.
Reviewer 3 Report
I checked your revise manuscript and I confirm your publication.
Author Response
Thank you very much for your meaningful comment. The manuscript is polished by the native speaker in the revised version.

Reviewer 4 Report
The reviewers' comments have been addressed.